# Optimization of Extraction Conditions for *Gracilaria gracilis* Extracts and Their Antioxidative Stability as Part of Microfiber Food Coating Additives

**DOI:** 10.3390/molecules25184060

**Published:** 2020-09-05

**Authors:** João Reboleira, Rui Ganhão, Susana Mendes, Pedro Adão, Mariana Andrade, Fernanda Vilarinho, Ana Sanches-Silva, Dora Sousa, Artur Mateus, Susana Bernardino

**Affiliations:** 1MARE-Marine and Environmental Sciences Center, ESTM, Politécnico de Leiria, 2520-641 Peniche, Portugal; rganhao@ipleiria.pt (R.G.); susana.mendes@ipleiria.pt (S.M.); pedro.adao@ipleiria.pt (P.A.); susana.bernardino@ipleiria.pt (S.B.); 2Department of Food and Nutrition, National Institute of Health Dr Ricardo Jorge (INSA), Avenida Padre Cruz, 1649-016 Lisbon, Portugal; mariana.andrade@insa.min-saude.pt (M.A.); fernanda.vilarinho@insa.min-saude.pt (F.V.); 3Center for Study in Animal Science (CECA), ICETA, University of Oporto, 4051-401 Oporto, Portugal; anateress@gmail.com; 4National Institute for Agricultural and Veterinary Research (INIAV), Vairão, 4485-655 Vila do Conde, Portugal; 5Centre for Rapid and Sustainable Product Development, Politécnico de Leiria, Zona Industrial, Rua de Portugal, 2430-028 Marinha Grande, Portugal; doralsousa@gmail.com (D.S.); artur.mateus@ipleiria.pt (A.M.)

**Keywords:** response surface methodology, *Rhodophyta*, electrospinning, active packaging, Box–Behnken design

## Abstract

Incorporation of antioxidant agents in edible films and packages often relies in the usage of essential oils and other concentrated hydrophobic liquids, with reliable increases in antimicrobial and antioxidant activities of the overall composite, but with less desirable synthetic sources and extraction methods. Hydroethanolic extracts of commercially-available red macroalgae *Gracilaria gracilis* were evaluated for their antioxidant potential and phenolic content, as part of the selection of algal biomass for the enrichment of thermoplastic film coatings. The extracts were obtained through use of solid-liquid extractions, over which yield, DPPH radical reduction capacity, total phenolic content, and FRAP activity assays were measured. Solid-to-liquid ratio, extraction time, and ethanol percentages were selected as independent variables, and response surface methodology (RSM) was then used to estimate the effect of each extraction condition on the tested bioactivities. These extracts were electrospun into polypropylene films and the antioxidant activity of these coatings was measured. Similar bioactivities were measured for both 100% ethanolic and aqueous extracts, revealing high viability in the application of both for antioxidant coating purposes, though activity losses as a result of the electrospinning process were above 60% in all cases.

## 1. Introduction

In order to achieve a desirable stability, meat and poultry products are kept in refrigerated storage and in protective packaging, often with use of modified atmosphere Common packaging atmospheres in western European markets can have up to 70% O_2_ and 30% CO_2_ [1]. High oxygen storage promotes higher concentrations of oxymyoglobin, responsible for an appealing red hue in most meat products but comes with considerable drawbacks. These include off-flavor producing by-products of lipid oxidation, and decreased tenderness due to protein oxidation. It is well known that elevated levels of carbon dioxide inhibit microbial growth, while elevated levels of oxygen prolong color stability. Higher carbon dioxide levels successfully inhibit microbial growth but have also been linked with degrading meat quality and production of off-flavors. Vacuum skin packaging fixes many of the problems associated with modified atmosphere packaging, but still has a noticeable effect on meat color. Lower oxygen levels have also been linked with less attractive color [2,3,4,5]. Lipid oxidation is not regarded as a limiting factor for shelf-life in aerobically preserved chilled meat, as it occurs at a slower rate than microbial degradation and discoloration. This is not always the case when dealing with modified atmosphere packaged meat, as the other deteriorative effects are suppressed. Poultry, which is richer in polyunsaturated fatty acids, is even more susceptible to lipid oxidation spoilage [1].

Active packaging is a recent technological development that has the potential of extending the shelf-life of meat and poultry. It has been defined as a material that performs a role other than serving as a simple inert barrier to the outside environment, often with the inclusion of antimicrobial and antioxidant agents directly into packaging [6]. A recent technological development that has been implemented to extend the shelf-life of meat and poultry, among other products. Several authors have reported increased food stability upon addition of antioxidant agents in active packaging. Common agents include butylated hydroxytoluene (BHT), butylated hydroxyanisole (BHA) and nisin. Pure standards of natural antioxidants have increasingly been used in alternative to these agents, and include α-tocopherol, caffeic acid, catechin, quercetin, and carvacrol [7].

Incorporation of antioxidant agents in edible films and packages often relies in the usage of essential oils and other concentrated hydrophobic liquids, with reliable increases in antimicrobial and antioxidant activities of the overall composite and consequent improvements in product shelf life [8,9]. These oils are often obtained from plant sources and the usage of organic solvents in their extraction has been criticized as to whether it constitutes a health hazard for consumers [7]. The use of water-soluble antioxidant agents is less common, due to lower activities and extraction yields. As such, using commonly available seaweeds as the source for both the main polymers and the supplementing antioxidant/antimicrobial agents in bioactive films can potentially reduce production costs and create a safer, more sustainable product [10].

Throughout time, marine algae have developed complex mechanisms to promote adaptation to external factors (e.g., UV radiation, salinity, and temperature stress) as well as to defend themselves from biological pressures such as competitors, grazers or parasites. To do so, these organisms divert resources into producing unique bioactive compounds that, when adequately processed have various applications for humankind. Adding the fact that approximately half of the global biodiversity exists in marine environments, the sea and its inhabitants provide a large source of novel, and potentially revolutionary bioactive compounds [10,11,12]. Red algae of the *Gracilaria* genus have gained increased commercial relevance in recent years as an aquacultured edible seaweed [13]. While already established as important sources of agar, recent studies have used *Gracilaria* spp. as a novel source of bioactive compounds with potential applications in the food, feed and pharmaceutical industries [13]. It has been found that *Gracilaria gracilis* extracts obtained with either organic solvents or water can exhibit antioxidant activity, although the overall activities were reported to be subject to seasonality [14]. Francavilla et al., 2013, performed several solid-liquid extractions on wild *G. gracilis* and obtained peak antioxidant response on the ethyl acetate fraction of summer and fall harvests. This trend was reversed in aqueous extractions, which had overall less bioactive potential but resulted in much higher extraction yield [14]. While some research on optimal aquaculture conditions for maximum bioactive yields is now being explored, the established demand for *G. gracilis* still lies in the production of agar, limiting the information on how much bioactive potential is retained in the cultured variants [15]. Regardless, the stability and availability of aquacultured *G. gracilis* makes it a compelling option when considering industrial integration [15].

Electrospinning is considered a viable new technique for immobilization of bioactive antioxidants for use in innovative packaging techniques. Electrospun fibers possess high surface-to-volume ratios and can be made from a limitless number of polymeric materials in order to suit a wide array of technological purposes [16]. The field of food grade electrospun polymers remains fairly open to innovation, but nylon 6,6; PVA (polyvinyl alcohol), PVDF (polyvinylidene difluoride), and PEO (polyethylene oxide) have seen wide use in food grade applications. PEO in particular has well documented properties, including high biocompatibility and availability, granting it popular use in food coating and electrospinning research [17,18].

Hydroethanolic extracts of commercially available red macroalgae *G. gracilis* were evaluated for their antioxidant potential and phenolic content, as part of the preliminary assays for the selection of algal biomass for the enrichment of thermoplastic films. Extracts that displayed the highest antioxidant potential were then processed through electrospinning into a microfiber coating, using high molecular weight PEO as the base polymer. Bioactive stability of the electrospun material was then evaluated through repeated antioxidant assays.

## 2. Results and Discussion

### 2.1. Selective Optimization of Extraction Conditions

Upon completion of all the required extractions and antioxidant assays, a set of equations representing the polynomial model for each response were obtained and were further used to construct the response surface graphs in Figure 1, Figure 2, Figure 3 and Figure 4, and to obtain the set of regression coefficients presented in Table 1.

The models have demonstrated that the antioxidant activity of the extracts is significant and above all other factors, influenced by solvent composition. Both 100% ethanolic and 100% aqueous extractions showed high activity for all responses, with the exception of the FRAP assay, which showed a much higher reducing potential on ethanolic extracts. Extraction yields were also significantly higher in aqueous extracts, given the amount of water-soluble algal components extracted, such as phycocolloids and proteins [14]. Prolonged extraction times did not significantly impact any of the evaluated responses, while SLR only had a noteworthy impact on extraction yield. Similar activities on aqueous and ethanolic compounds were observed, considering the main source of antioxidant activity from *G. gracilis* likely comes from phenolic compounds, a class of organic molecule that has very comparable extraction yields between these two solvents. The cold aqueous extracts may also contain agaropectin, which is the sulfated portion of the agar found in *Gracilaria.* spp. [19]. Agaropectin is water-soluble at room temperature, whereas agarose typically requires high temperatures for optimal extraction yields [20]. Sulfated galactan phycocolloids have been reported to exhibit some antioxidant and biological properties, which can partly explain the observed antioxidant activity of the 100% aqueous extract [21,22,23]. Furthermore, red macroalgae contain water-soluble phycoboliproteins which also carry significant antioxidant potential [14,24,25]. The presence of phycobiliproteins may also contribute to the overall antioxidant potential of the aqueous extract per gram of dry extract. The quantification of phycobiliproteins lay beyond the scope of this project, but has been previously quantified for wild *G. gracilis*, and appears less subjected to seasonal variation as the overall antioxidant activities and protein content [26,27,28].

High antioxidant responses in both aqueous and ethanolic extracts is not commonly observed in literature for *G. gracilis*, leading to the belief that little potential lies in crude water extracts [13,14,28,29]. From the published content that we gathered, this is a matter of experimental design, as most published content that screens *G. gracilis* for antioxidant potential using multiple solvents in solid-liquid extractions, does so in sequential fractionated procedures. These often involves washing of a single portion of biomass with every solvent, starting from the least polar. The result of this is an aqueous extract that has been depleted of most of its phenolic content by slightly less polar solvents (e.g., methanol and ethanol) and thus harboring less antioxidant potential. Simultaneously, there are numerous studies that focused on the extraction of sulphated polysaccharides from *Gracilaria* spp., and successfully verified high antioxidant activities in aqueous fractions [14,21,22].

While the results fall short of providing an accurate model of the dynamics of extraction, they have supplied useful information on how to tailor extraction conditions for dried *G. gracilis* when attempting to make use of its bioactivities. They also encourage the consideration of more than one extract on further testing (e.g., 100% aqueous and 100% ethanolic), due to similar antioxidant potentials in two of the three tests, and the very high yields recovered in aqueous extracts. Nevertheless, a model that allows a flexible array of choices can be most useful. As an example, the aqueous extracts showing antioxidant potential comparable to those found on ethanolic extracts, while having greater yield, makes them a very desirable choice when considering the hazards and environmental impact associated with the use of organic solvents. Even the presence of phycocolloids in the aqueous extracts may have benefits on the electrospinning process, or on the physical properties of the coating itself [30].

### 2.2. Results of Electrospinning Technique

For the first set of coatings produced, three extract formulations were considered. These were obtained following 10 min of extraction at a 1:25 SLR, and with either water, 50% ethanol or 100% ethanol used as solvents. All other details were according to the general method described previously. Although the RSM model strongly discouraged the use of 50% ethanolic solutions as the extraction solvent, its inclusion in the coating tests was deemed valuable for the potential information it could provide, as extract composition could play a critical part in the length and distribution of the microfibers. Photos of optical microscopy of these coatings can be seen in Figure 5, Figure 6 and Figure 7.

The photos show a noticeable reduction in fiber length in extracts which have used ethanol as their extraction solvent. A certain degree of variability in microscopic scale structure was to be expected, as the extract is a major component of the electrospinning solution. It is assumed that the high amount of phycocolloids present in the aqueous extract, most likely water-soluble agaropectins in the case of *G. gracilis*, in the aqueous extracts contributes to the stability of the fibrous structure. This had clear macroscopic ramifications, as could be verified when attempting to peel off the coatings from the polypropylene film for the purpose of re-evaluating the antioxidant potentials. Aqueous and 50% ethanolic extract coatings were easily removed as a single layer, while 100% ethanolic extract coatings had much lower structural integrity and had to be scrapped off the film as a sticky cohesive powder. These different behaviors will likely have important effects on mass transfer when in contact with food, particularly high moisture content food, and may yet reveal to be useful or detrimental to the intended application of the coatings. Limited mass transfer is often preferred in antioxidant applications, as to lengthen the useful time of the active agents, and to avoid pro-oxidative effects due to high concentrations of reducing agents in the food matrix [3,31,32].

### 2.3. Antioxidant Stability of Electrospun Coatings

Table 2 shows the percentual loss in antioxidant activity per gram of extract, comparing the original dried extracts (prior to electrospinning) to the electrospun coating that was scraped off the final films. Slight deviations from the standard electrospinning protocol happened during the production of some of these coatings and are listed in the description of each.

The results show a loss of antioxidant activity higher than 60% for all extract coatings in all the measured responses, the most likely cause for this being the degradation of reactive antioxidant compounds due to rapid oxidation and prolonged exposure to air. This process is likely to degrade the highly reactive phenolic species present in the extracts, and responsible for part of their pre-electrospinning activity [29]. What residual activity remains can thus be attributed to the remaining of phenolic compounds at time of measure, as well as the more resilient galactan phycocolloids. The relative stability of these compounds means that their antioxidant potential can be observed well after the early degradation of the most reactive electron scavengers [21,22,23]. Further tests will be required in order to understand the biochemical changes that these natural products undertake when subjected to the electrospinning process, as well as how the antioxidant potential of these films’ changes through time. A delayed, low-intensity effect is in fact the desired means of action for most antioxidant agents in active packaging applications, and additional studies will be required to assess the long-term antioxidant activity.

## 3. Materials and Methods

### 3.1. Materials

Dried *G. gracilis* samples were purchased from ALGAplus (Ílhavo, Portugal) and subsequently ground to a rough powder using a kitchen blender. 2,2-diphenyl-1-picrylhydrazyl (DPPH), 6-hydroxy-2,5,7,8-tetramethylchroman-2-carboxylic acid (Trolox), and 2,4,6-Tris(2-pyridyl)-s-triazine (TPTZ) were all purchased from Sigma-Aldrich Chemical Co. (St. Louis, MO, USA). Folin-Ciocateu reagent was purchased from PanReac Química SLU (Barcelona, Spain). Analytical grade gallic acid and ascorbic acid were purchased from Merck KGgA (Darmstadt, Germany). All remaining chemicals used were of analytical grade.

### 3.2. Selective Optimization of Extraction Conditions

*G. gracilis* extracts were obtained through use of solid-liquid extractions, over which yield, DPPH radical reduction capacity, total phenolic content, and FRAP activity assays were measured [4,5]. Solid-to-liquid ratio (SLR), extraction time, and ethanol to water ratio were selected as independent variables with experimental ranges and configurations obtained using a Box-Behnken design with three factors, resulting in 15 experimental conditions. The levels of independent variables used in this design are listed in Table 3. Extraction duplicates were used.

Conditions ranged from 10 to 180 min of extraction time, 1:5 to 1:25 (grams of algal mass to milliliter of solvent) of SLR, and 0 to 100% ethanol in distilled water ratio. All extractions took place inside 50 mL plastic centrifuge tubes, and were homogenized for the duration of the extraction using a laboratory see-saw rocker (Stuart SSL4, Cole-Parmer, Staffordshire, UK) at 70 rev./min. Response surface methodology (RSM) was then used to model the antioxidant potential of the different extracts. All results were considered statistically significant at the 5% level. All calculations were performed using the software Statistica v12 (StatSoft Inc., Minneapolis, MN, USA).

### 3.3. Antioxidant Activity Assays

The DPPH radical reduction capacity assay was adapted from Ye et al. (2008) with modifications to the sample dilution rate. Two microliters of sample material were added to 198 microliters of 0.1 mm ethanolic DPPH. Mixture discoloration was measured at 517 nm after a 30 min incubation at 30 °C. The percentage of scavenged DPPH was measured and compared to a set of trolox standards, with concentrations ranging from 1 to 0.1 mg/mL. Total phenolic content was estimated using Folin-Ciocalteu reagent, with a procedure adapted from Swain and Hillis (1959). Two microliters of sample material were added to 168 microliters of 6.25% (*v/v*) Folin-Ciocalteu solution, after which 30 microliters of a 20% (*m/v*) sodium carbonate solution were added. The mixture was incubated for a period of 1 h at 30 °C, and its absorbance at 755 nm was read. Gallic acid standards were used, with concentrations ranging from 1 to 0.01 mg/mL. FRAP activity assay was performed according to Benzie and Strain (1996), with slight modifications to sample dilution rates. 195 μL of ferric TPTZ, along with 5 μL of either sample or ascorbic acid standard were incubated for 30 min at 30 °C. The concentrations of the latter ranged from 1000 to 20 µM. A minimum of three independent assays were performed for each extraction condition or film suspension tested.

### 3.4. Electrospinning Technique

Prior to the production of the microfibers, the dried seaweed extracts were suspended in 50% ethanolic 1% PEO (MW = 900,000 g/mol) solutions, at an 8 mg/mL concentration. Solvent composition was selected in order to completely dissolve all extracts tested, regardless of extraction condition. For the production of the extract-enriched PEO fibers, a variable voltage electrospinning device was used. Polymer solutions were introduced through a 5 mL plastic syringe, connected to a gauge 22 stainless steel needle. The microfibers were collected in PP (polypropylene) sheets, which covered the 10 by 10 cm copper collector. Experiments were carried out at room temperature (21 °C) with 45% humidity. Flow rate was kept at 0.4 mL/h, and each fiber coating was spun over a period of approximately 7 h, at 12 kV, with a 10 cm needle tip to collector distance.

Upon completion of the electrospinning process, the coatings were inspected under optical microscope, at 100× amplification, so fiber length and homogeneity could be determined.

### 3.5. Antioxidant Stability of Electrospun Coatings

After producing the first successful electrospun coatings, these were subjected to a new round of antioxidant assays, similar to those used in the extract optimization step. DPPH radical reduction capacity, total phenolic content and FRAP activity assays were carried out on the coating material that was scraped off the polypropylene film with a steel spatula. The seaweed extract content of the final coatings was approximately 44% of its dry weight, based on the PEO content of the electrospinning solution and the assumption of complete water evaporation by the end of the electrospinning process. The coatings were kept in a desiccator for 72 h after they were applied, and water content was considered negligible by time of testing. Scraped films of 1% PEO (MW = 900,000 g/mol) and aqueous *G. gracilis* extracts at 8 mg/mL can be seen in Figure 8, halfway through the process. The same solution was tested three times, producing three identically coated films.

## 4. Conclusions

The work presented is meant as the starting point for new advances on the use of marine natural products for active packaging applications through the use of electrospinning technologies. It has revealed a series of drawbacks regarding the direct use of macroalgae extracts in commonly used electrospinning settings, that can be resolved with further work, helping the continuous effort of employing low-value raw material in new and sustainable applications.

While the extract optimization ultimately aims to restrict the number of tested conditions in future trials, the results obtained show that there is a potential applicability of either aqueous or ethanolic extracts of *G. gracilis* in the development of active packaging materials. The extraction methodologies are relatively straightforward and employ solvents of low toxicity and environmental impact. The aqueous extraction methodology is particularly attractive since the resulting extracts yield antioxidant responses identical to the ethanolic extracts. In addition, the fibers spun from the aqueous extracts exhibited greater structural integrity than those obtained from the ethanolic extracts, making the aqueous extracts more suited for the electrospinning process.

While there was a significant loss of antioxidant activity measurable by the DPPH, FRAP, and TPC assays, it should be noted that this activity can be attributed mostly to highly reactive phenolic species in the presence of an excess of strong oxidants. Long term antioxidant activity is desirable, and the galactan phycocolloids present in the aqueous extracts can provide this necessary long-term activity. Further work needs to take place in order to understand how these different extracts interact with different electrospinning configurations and ultimately, the food matrix itself.

## Figures and Tables

**Figure 1 molecules-25-04060-f001:**
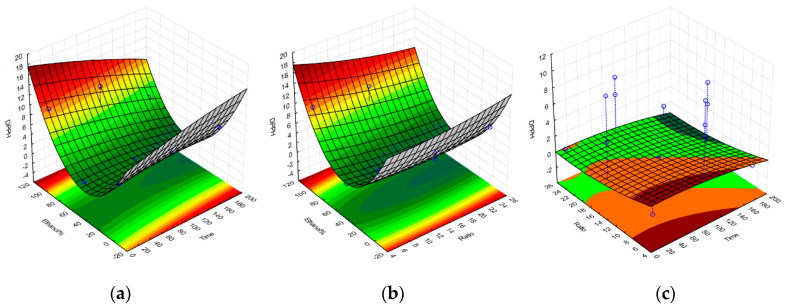
Response surface graphs for the model of the *G. gracilis* extractions, obtained for the 2,2-diphenyl-1-picrylhydrazyl (DPPH) response upon variation of ethanol% and extraction time (**a**), ethanol% and solid-to-liquid ratio (**b**), and solid-to-liquid ratio and extraction time (**c**). DPPH radical reduction activity expressed in mg of trolox equivalents per g of dried extract. Time is expressed in minutes, and solid-to-liquid ratio (SLR) (here displayed as Ratio) as solvent volume (mL) per one gram of dried seaweed.

**Figure 2 molecules-25-04060-f002:**
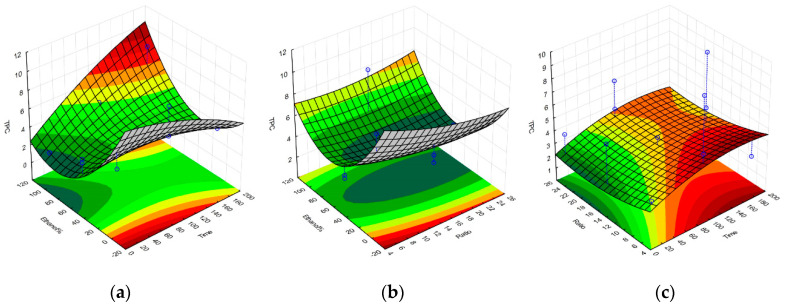
Response surface graphs for the model of the *G. gracilis* extractions, obtained for the TPC response upon variation of ethanol% and extraction time (**a**), ethanol% and solid-to-liquid ratio (**b**), and solid-to-liquid ratio and extraction time (**c**). Total phenolic content expressed in mg of gallic acid equivalents per g of dried extract. Time is expressed in minutes, and SLR (here displayed as Ratio) as solvent volume (mL) per one gram of dried seaweed.

**Figure 3 molecules-25-04060-f003:**
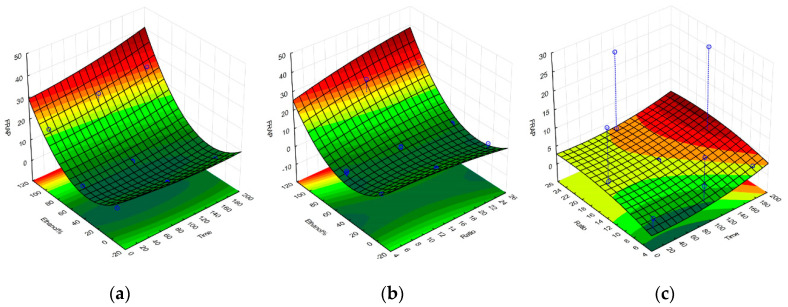
Response surface graphs for the model of the *G. gracilis* extractions, obtained for the FRAP response upon variation of ethanol% and extraction time (**a**), ethanol% and solid-to-liquid ratio (**b**), and solid-to-liquid ratio and extraction time (**c**). FRAP activity expressed in µmol of ascorbic acid equivalents per g of dried extract. Time is expressed in minutes, and SLR (here displayed as Ratio) as solvent volume (mL) per one gram of dried seaweed.

**Figure 4 molecules-25-04060-f004:**
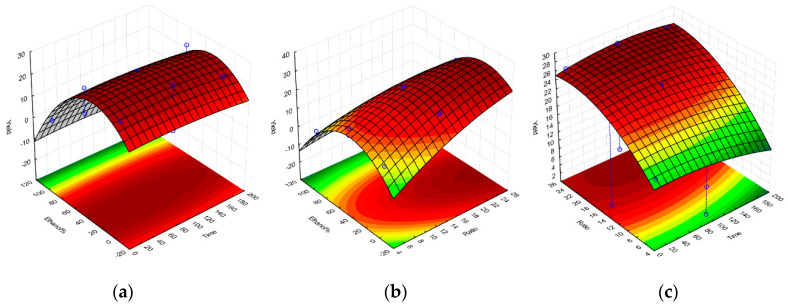
Response surface graphs for the model of the *G. gracilis* extractions, obtained for the Yield response upon variation of ethanol% and extraction time (**a**), ethanol% and solid-to-liquid ratio (**b**), and solid-to-liquid ratio and extraction time (**c**). Yield is expressed as a percentage of dry weight. Time is expressed in minutes, and SLR (here displayed as Ratio) as solvent volume (mL) per one gram of dried seaweed.

**Figure 5 molecules-25-04060-f005:**
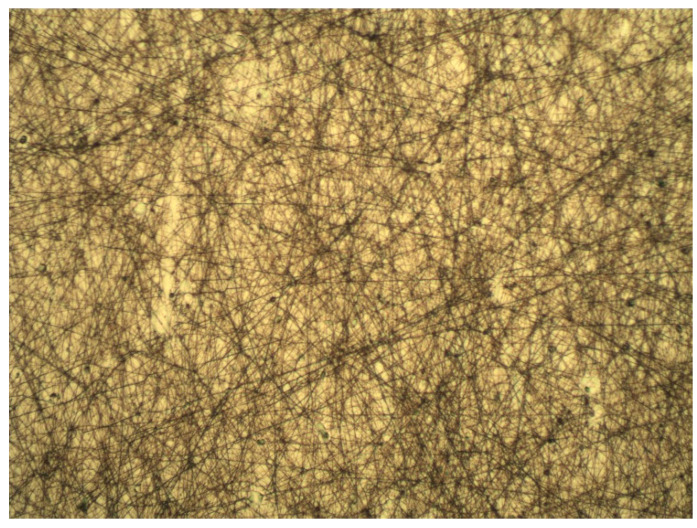
Optical microscope image of the electrospun aqueous extracts. 100× amplification.

**Figure 6 molecules-25-04060-f006:**
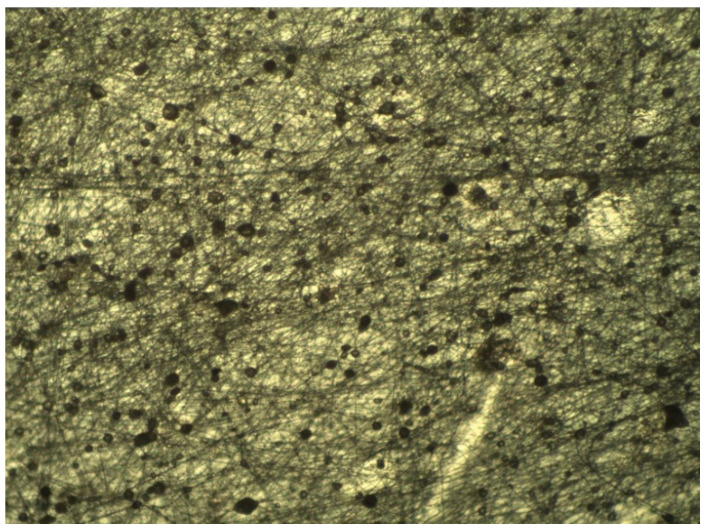
Optical microscope image of the electrospun 50% ethanol extracts. 100× amplification.

**Figure 7 molecules-25-04060-f007:**
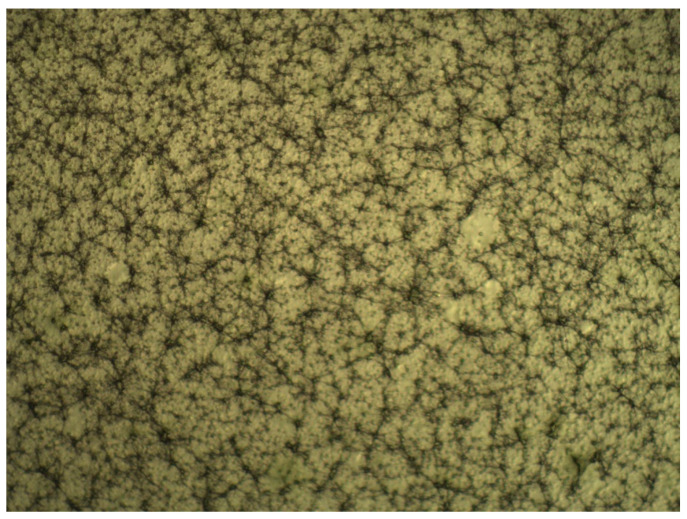
Optical microscope image of the electrospun 100% ethanol extracts. 100× amplification.

**Figure 8 molecules-25-04060-f008:**
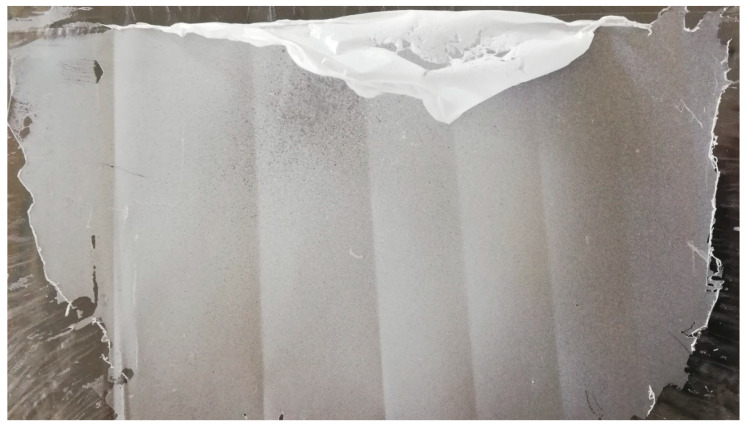
Polypropylene film with a partially displaced PEO (polyethylene oxide) and aqueous *G. gracilis* extract coating, for the purpose of re-evaluation of antioxidant activities.

**Table 1 molecules-25-04060-t001:** Regression coefficients for the developed response surface model. *, ** and *** indicate parameter significance to the corresponding response, with *p* < 0.05; 0.01 and 0.001 respectively.

Regression Coefficient β	DPPH	TPC	FRAP	Yield
**Intercept**	9.057	6.642	−1.220	5.077
**Linear**				
Time	0.43	0.006	−0.256	0.012
SLR	−0.200	−0.827	0.156	1.392 **
Ethanol%	−1.744 **	−2.917 *	−1.594 **	1.452 ***
**Quadratic**				
Time^2^	−0.152	−0.448	0.208	−0.166
SLR^2^	0.298	0.509	−0.310	−0.717 *
Ethanol%^2^	2.768 ***	1.706	1.766 **	−1.702 ***
**CrossProduct**				
Time*SLR	−0.005	−0.018	−0.014	0.137
Time*Ethanol%	−0.645 *	1.137	0.326	0.014
SLR*Ethanol%	−0.463	0.366	0.556	−0.653 **
R^2^	0.861	0.270	0.890	0.972
*p* value of lack of fit	0.0011	0.0436	0.0018	0.0090
*p* value of the models	0.0091	0.3207	0.0052	0.0002

**Table 2 molecules-25-04060-t002:** Percentage loss of antioxidant activity per gram of dried extract subject to electrospinning. * No DPPH radical reduction was detected using the available instrumentation. Values displayed correspond to the average of three samples assayed in triplicate, ± standard deviation.

	DPPH	TPC	FRAP
**Aqueous**; 0.3 mL/h, 14 kV, 10 cm dist. 47% RH; 22 °C	78.61 ± 6.6	63.37 ± 4.2	88.89 ± 6.9
**50% Ethanolic**; 0.4 mL/h, 11 kV; 11 cm dist., 46% RH, 21 °C	100 *	78.61 ± 2.3	80.36 ± 1.1
**100% Ethanolic**; 0.5 mL/h; 11 kV; 11 cm dist., 40% RH, 21 °C	76.27 ± 2.8	90.93 ± 1.1	89.9 ± 3.8

**Table 3 molecules-25-04060-t003:** Box-Behnken design layout and legend used in the extraction optimization procedure for *Gracilaria gracilis*. Total number of unique experimental conditions = 13, total number of trials = 15.

X1	X2	X3	Ethanol%	Time (min)	SLR (g/mL)
−1	−1	0	0	10	1/55
−1	0	−1	0	95	1/10
−1	0	1	0	95	1/100
−1	1	0	0	180	1/55
1	−1	0	100	10	1/55
1	0	−1	100	95	1/10
1	0	1	100	95	1/100
1	1	0	100	180	1/55
0	−1	−1	50	10	1/10
0	−1	1	50	10	1/100
0	0	0	50	95	1/55
0	0	0	50	95	1/55
0	0	0	50	95	1/55
0	1	−1	50	180	1/10
0	1	1	50	180	1/100

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
