# Peer review of "Optimization of Extraction Conditions for Gracilaria gracilis Extracts and Their Antioxidative Stability as Part of Microfiber Food Coating Additives"

_molecules, 2020, doi:10.3390/molecules25184060_

Round 1

Reviewer 1 Report

Reboleira et al reported a new technology and optimized GG extracts as well as analyze the antioxidative stability as microfiber food coating additives. This study is meaningful and applicable.  But the research is not good design, not good writing and the contents were very confusing. I suggested authors rearrange the study logic and rewrite the paper, then we can review again based on the new edition. There are some issues suggested to be clarified.

  1. Line39-42, the contents are too verbose and lacks of key points.
  2. Line45-53, the contents are not related to research aim.
  3. In the part of introduction, it lacks of Gracilaria gracilis information, such as species, output all of world, the differences among other seaweeds, and so on.
  4. I suggest authors rewrite the part of introduction to supply more information and aim about your research.
  5. Line 96, G. gracilis should be explained when it is firstly shown in the paper.
  6. Line 103, seaweed extracts should be replaced by “G. gracilis extracts”
  7. Line 107, why is it 15 experimental conditions? Usually, it should be 17 conditions for getting good parallel data.
  8. Line 120, 2 uL should be replaced by Two microliters.
  9. Line123 and Line 125, there should be space between 0,1 and 6,25%
  10. Line184-198, it lacks of explaining figure2-5 and response surface data.
  11. Figure 6-8 should be made in one figure.
  12. Table 3 should be explained more detail and concise, the data means scavenging rate (%) I think, but there is no this information.
  13. The conclusion should be rewritten.

Author Response

Thank you very much for your thorough review of our submission. Below you will find our replies to your questions, as well as the tracked changes in the resubmitted manuscript.

1. Line39-42, the contents are too verbose and lacks of key points.

A slight restructuring of the first paragraph was made to make it easier to understand. Please see the tracked changes in the revised manuscript.

2. Line45-53, the contents are not related to research aim.

While the matter of modified atmosphere is not directly addressed in the methodology, the current meat preservation techniques are based on modified atmosphere and on the effect it has on meat properties. Specifically, the alterations in meat quality detailed in these lines are very much related to the action of oxygen and oxygen radicals, countered by antioxidants. As such, we remain convinced that this piece of content is relevant in the context discussing innovation of any poultry packaging technology.

3. In the part of introduction, it lacks of Gracilaria gracilis information, such as species, output all of world, the differences among other seaweeds, and so on.

A section with information relative to G. gracilis, its uses and potential as source of bioactivity was included in the introduction as recommended between lines  88 and 101.

4. I suggest authors rewrite the part of introduction to supply more information and aim about your research.

See answer to point 1.

5. Line 96, G. gracilis should be explained when it is firstly shown in the paper.

See answer to point 3.

6. Line 103, seaweed extracts should be replaced by “G. gracilis extracts”

It has been replaced, now in line 125.

7. Line 107, why is it 15 experimental conditions? Usually, it should be 17 conditions for getting good parallel data.

Considering the limitations of biomass we had, we considered the use of an efficient Box-Behnken design, as to reduce our number of experimental conditions, yet retain statistical validity. A standard Box-Behnken experimental design with 3 factors results in 15 conditions and seems to be widely accepted. A similar design was recently published here: https://www.mdpi.com/2076-3417/10/15/5304

8. Line 120, 2 uL should be replaced by Two microliters.

9. Line123 and Line 125, there should be space between 0,1 and 6,25%

Corrections have been made on the appropriate section. Please see the tracked changes in the revised manuscript.

10. Line184-198, it lacks of explaining figure2-5 and response surface data.

This section has been modified for clarity. Please see the tracked changes in the revised manuscript.

11. Figure 6-8 should be made in one figure.

Upon attempting to join all figures together, we found that an accurate description or legend for the combined figure was confusing due to its length. As such, we retained the three separate figures for clarity. If you still find this unacceptable, please let us know and we will attempt a different solution.

12. Table 3 should be explained more detail and concise, the data means scavenging rate (%) I think, but there is no this information.

Table 3 presents a percentage loss of antioxidant activity, using the unprocessed extracts (obtained after the RSM optimization) as the 100% reference, and displaying how much of that activity was lost after electrospinning. A slight modification on the table’s description was added on lines 294 to 296.

13. The conclusion should be rewritten.

The conclusion was modified to better reflect the relevant conclusions of this work. Please see the tracked changes in the revised manuscript.

I hope that covers everything you have mentioned. Again, thank you for your thorough review, and for helping us improve our content.

Best regards,

João Reboleira

Reviewer 2 Report

Major points:

  1. Authors should measure phycobiliproteins.
  2. In the introduction Authors should discuss previous results on antioxidant activities of Gracilaria gracilis extracts and in particular the reasons of variability (season…)
  3. Methods and results on antioxidant stability should be clarified, i.e:

“ An approximate concentration of 8mg/mL of seaweed extract in the electrospinning solution was considered when calculating specific activity, and it was assumed all water had evaporated off the coatings by the time they were tested, which was around 72 hours after the production of the film coatings. “

“Considering that the coatings were resuspended in a 50% ethanolic, similar to the electrospinning solution they were made from, prior to the antioxidant potential assays, the most likely cause for the drop in activity is the degradation of the antioxidant compounds themselves, possibly due to rapid oxidation and prolonged exposure to air. Further tests will be required in order to understand the biochemical changes that these natural products undertake when subjected to the electrospinning process, as well as how the antioxidant potential of these films changes through time. A delayed, low-intensity effect is in fact the desired means of action for most antioxidant agents in active packaging applications, and it remains to be seen if these coatings can still supply that.”

  1. In the results/discussion section Authors should better discuss antioxidant activities and other extraction methods used for algae from literature and compare to their results.

Minor points:

  1. Check tables’ and figures’ legends: they should be self-explanatory
  2. Check for refusal from format downloaded from journal page: “ This section may be divided by subheadings. It should provide a concise and precise description of the experimental results, their interpretation as well as the experimental conclusions that can be drawn.”

Author Response

Thank you very much for your in-depth and exceptionally thorough review. Below you will find our replies to your major points, as well as a revised manuscript with all the tracked changes.

1. Authors should measure phycobiliproteins.

The quantification of PBPs was considered outside the initial scope of this project, and thus was not undertaken. We do recognize the relevance of this information when addressing the antioxidant potential of red algae, and thus included additional information on the PBP content and stability of Gracilaria species. You can find it in lines 232 to 236 of the manuscript with tracked changes.

2. In the introduction Authors should discuss previous results on antioxidant activities of Gracilaria gracilis extracts and in particular the reasons of variability (season…)

An additional portion of content was added to the introduction related to the bioactive potential of G. gracilis and its seasonality, lines 88 to 101. Please see the tracked changes in the revised manuscript.

3. Methods and results on antioxidant stability should be clarified, i.e:

“ An approximate concentration of 8mg/mL of seaweed extract in the electrospinning solution was considered when calculating specific activity, and it was assumed all water had evaporated off the coatings by the time they were tested, which was around 72 hours after the production of the film coatings. “

“Considering that the coatings were resuspended in a 50% ethanolic, similar to the electrospinning solution they were made from, prior to the antioxidant potential assays, the most likely cause for the drop in activity is the degradation of the antioxidant compounds themselves, possibly due to rapid oxidation and prolonged exposure to air. Further tests will be required in order to understand the biochemical changes that these natural products undertake when subjected to the electrospinning process, as well as how the antioxidant potential of these films changes through time. A delayed, low-intensity effect is in fact the desired means of action for most antioxidant agents in active packaging applications, and it remains to be seen if these coatings can still supply that.”

The parts mentioned have been thoroughly revised and can be found in lines 172 to 176 and 303 to 314 respectively. Please see the tracked changes in the revised manuscript.

4. In the results/discussion section Authors should better discuss antioxidant activities and other extraction methods used for algae from literature and compare to their results.

While the authors agree on the importance of comparisons when adequate, we found it difficult to apply said comparisons in our study. This was mostly because of the restraints on the extraction methods applied, allowing their compatibility with both electrospinning and food-grade applications. Some comparisons with similar solid-liquid extraction conditions were added whenever we thought were applicable, and an overall comparison to extracts from G. gracilis was included in lines 237 to 246. Please see the tracked changes in the revised manuscript.

A comparison with other seaweed species was not within the scope of this project, as it aims to optimize the extraction of G. gracillis bioactives, the potential of which is described in the introduction. Future projects may consider the inclusion of other species and their subsequent discrimination via highest bioactive potential.

I hope that you find our revisions satisfactory. Thank you once again for your thorough review and for helping us improve our content.

Best regards,

João Reboleira

Round 2

Reviewer 1 Report

All issues were addressed.

Reviewer 2 Report

.